# Masked Capsule Autoencoders

**Miles Everett**                                                                 *miles.everett@abdn.ac.uk*
*Department of Computing Science*
*University of Aberdeen, UK*

**Mingjun Zhong**                                                              *mingjun.zhong@abdn.ac.uk*
*Department of Computing Science*
*University of Aberdeen, UK*

**Georgios Leontidis**                                                    *georgios.leontidis@abdn.ac.uk*
*Interdisciplinary Institute*
*Department of Computing Science*
*University of Aberdeen, UK*

**Reviewed on OpenReview:** *https://openreview.net/forum?id=JHxrhOOW1j*

## Abstract

We propose Masked Capsule Autoencoders (MCAE), the first Capsule Network that utilises pretraining in a modern self-supervised paradigm, specifically the masked image modelling framework. Capsule Networks have emerged as a powerful alternative to Convolutional Neural Networks (CNNs). They have shown favourable properties when compared to Vision Transformers (ViT), but have struggled to effectively learn when presented with more complex data. This has led to Capsule Network models that do not scale to modern tasks. Our proposed MCAE model alleviates this issue by reformulating the Capsule Network to use masked image modelling as a pretraining stage before finetuning in a supervised manner. Across several experiments and ablations studies we demonstrate that similarly to CNNs and ViTs, Capsule Networks can also benefit from self-supervised pretraining, paving the way for further advancements in this neural network domain. For instance, by pretraining on the Imagenette dataset—consisting of 10 classes of Imagenet-sized images—we achieve state-of-the-art results for Capsule Networks, demonstrating a 9% improvement compared to our baseline model. Thus, we propose that Capsule Networks benefit from and should be trained within a masked image modelling framework, using a novel capsule decoder, to enhance a Capsule Network's performance on realistically sized images.

## 1 Introduction

Capsule Networks are an evolution of Convolutional Neural Networks (CNNs) which remove pooling operations and replace scalar neurons with a fixed number of vector or matrix representations known as capsules at each location in the feature map. At each location, there will be multiple capsules, each theoretically representing a different concept. Each of these capsules has a corresponding activation value between 0 and 1. The activation value measures the probability (from 0 to 1) that the concept represented by the capsule exists. Capsule Networks have shown promising signs, such as being naturally strong in invariant and equivariant tasks (Sabour et al., 2017; Hinton et al., 2018; De Sousa Ribeiro et al., 2020; Hahn et al., 2019; Ribeiro et al., 2020; 2022) while having low parameter counts, but have yet to scale to more complex datasets with realistic resolutions that CNNs and Vision Transformers (ViTs) are typically benchmarked on.

Masked Image Modelling (MIM) is a Self Supervised Learning (SSL) technique with roots in language modelling (Devlin et al., 2018). In language modelling, words are removed from passages of text, the network is then trained to predict the correct words to fill in the gaps. This technique can be extended to

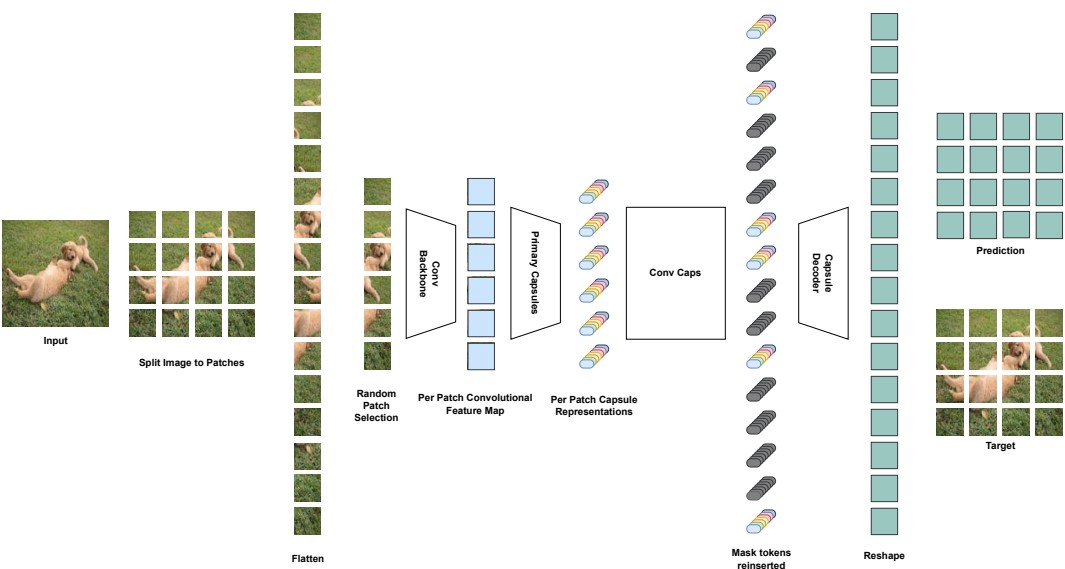

Figure 1: Our Masked Capsule Autoencoder architecture. During pretraining we randomly select a number of patches from the original image to be processed. The Capsule Network will then create a representation for each patch. Masked patch capsule representations are then re-added before the capsule decoder, where the unmasked capsules can contribute to the masked positions, which are finally decoded by a single linear layer to the original patch dimensions. The pretraining objective is the mean squared error between the reconstructed patches and the target patches. The dog image used is sourced from the Imagewoof validation set (Howard, 2019b).

image modelling by splitting an image into equal regions called patches, randomly removing some of these patches and then requiring the network to predict the pixel values of the removed patches. This has been shown to require the network to have an improved world model in both Vision Transformers (ViTs) (He et al., 2021) and CNNs (Woo et al., 2023), which is strong enough to reconstruct occluded areas from the remaining visible areas. Combining this technique with supervised finetuning, accuracy can be significantly improved compared to not using any pretraining (He et al., 2021; Woo et al., 2023).

We propose incorporating MIM pretraining into the training process of Capsule Networks to address its shortcomings. By doing so, Capsule Networks can learn more refined representations for each region of the image, leading to improved reconstruction accuracy. These refined local representations can subsequently be employed during the finetuning phase to effectively activate the appropriate global class capsules, which are introduced after pretraining. These proposals are based on the observed significant improvements in CNNs and ViTs, particularly in terms of better training regimes with minimal architectural changes. Instead of solely focusing on enhancing the routing algorithm, we demonstrate that Capsule Networks can be improved through training modifications.

The main contributions of this work can be summarised as follows:

1. We propose a novel adaption of Capsule Networks to accommodate masked image modelling.

2. We demonstrate that the classification accuracy of a Capsule Network improves when self-supervised pretraining is introduced.

3. We achieve state-of-the-art performance on multiple benchmark datasets for Capsule Networks, including those with realistically sized images where Capsule Networks typically perform poorly.

4. We implement a fully capsule-based decoder layer, replacing the CNN decoders commonly used for reconstruction tasks in Capsule Networks, ensuring that our proposed MCAE model does not require a handcrafted decoder.

5. We conduct the first investigation into the use of ViTs as a replacement for the traditional convolutional stem.

The rest of this paper presents the necessary background on Capsule Networks, highlighting previous research that has inspired the work presented here. We then formally define our new self-supervised capsule formulation called Masked Capsule Autoencoders and present several experiments and ablation studies on benchmark datasets. We conclude the paper by highlighting the main advantages of our new methods, with some key takeaway messages and some future directions that could further support the future developments of large-scale self-supervised Capsule Network models.

While interest in Capsule Networks may have waned in recent years, we argue that this is largely due to research being overly focused on routing algorithms, leading to incremental improvements on toy datasets without addressing fundamental challenges such as handling complex images. This work represents a significant departure from this trend by proposing a novel training regime for Capsule Networks, analogous to the advancements seen in ViTs and CNNs. We posit that this approach could reignite interest in capsule networks and encourage researchers to explore alternative training paradigms beyond the standard supervised learning framework.

## 2 Related Works

### 2.1 Capsule Networks

**Capsule Networks** are a variation of CNNs, which replace scalar neurons with vector or matrix capsules and construct a parse tree, representing part-whole relationships within the network. Each type of capsule in a layer of capsules can be thought of as representing a specific concept at the current level of the parse tree which is part of a bigger concept. Capsules in deeper layers are closer to the final class label than capsules in shallower layers. The capsules in the lowest layer, known as primary capsules, detect the most basic visual features. These basic features serve as building blocks that can be combined to form any of the final object classes, forming the foundation of the parse tree. Capsules in lower layers decide their contribution to capsules in higher layers through a process called routing.

**Capsule Routing**, in brief, is a non-linear, cluster-like process that takes place between adjacent capsule layers. This part of the network has been the predominant research focus for state-of-the-art Capsule Networks, to find better or more efficient methods of finding ways to decide the contribution of lower capsules to higher capsules. In brief, the purpose of capsule routing is to assign *part* capsules $i = 1, \ldots, N$ in layer $\ell$ to *object* capsules $j = 1, \ldots, M$ in layer $\ell+1$, by adjusting coupling coefficients $\boldsymbol{\gamma} \in \mathbb{R}^{N \times M}$ iteratively, where $0 \leq \gamma_{ij} \leq 1$. These coupling coefficients have similarities to an attention matrix (Vaswani et al., 2017) which modulates the outputs as a weighted average of the inputs. For more information on the numerous routing algorithms proposed for Capsule Networks, please see here (Ribeiro et al., 2022).

**Dynamic Routing Capsule Networks** are the original Capsule Network architecture, as described in (Sabour et al., 2017). DR Caps employs a technique called dynamic routing to iteratively refine the connections between capsules. This approach introduces the concept of coupling coefficients which represent the strength of each connection and updates them using a softmax function to ensure that each capsule in a lower layer must split its contribution amongst capsules that it deems relevant in the higher layer. The update process relies on agreement values calculated as the dot product between a lower-level capsule's output and a predicted output from a higher-level capsule. After a pre-determined number of iterations, the activation of each higher-level capsule is calculated as the weighted sum of the lower-level capsule activations, where the weights are the final coupling coefficients.

**Self-Routing Capsule Networks** (SR-CapsNet) Hahn et al. (2019) address the computational burden of iterative routing algorithms in traditional Capsule Networks by introducing an efficient, independent routing mechanism. Unlike conventional routing-by-agreement approaches, SR-CapsNet employs a dedicated routing network for each capsule to compute its coupling coefficients directly, rather than relying upon iterative agreement methods, drawing inspiration from mixture of experts networks Masoudnia & Ebrahimpour (2014).

Let us formally define a layer in SR-CapsNet. Given an input feature map with batch size $b$, the activation tensor $\mathbf{a} \in \mathbb{R}^{b \times A \times h \times w}$ represents capsule activations across a feature map, where $A$ is the number of input capsule types and $h, w$ are spatial dimensions of the feature map. Each capsule also maintains a pose matrix $\mathbf{pose} \in \mathbb{R}^{b \times AC \times h \times w}$, where $C$ represents the dimensionality of each capsule's pose vector.

The self-routing mechanism is parameterised by two learnable weight matrices: $\mathbf{W}^{route} \in \mathbb{R}^{kkA \times B \times C}$ for computing routing coefficients and $\mathbf{W}^{pose} \in \mathbb{R}^{kkA \times BD \times C}$ for pose transformation, where $k$ is the convolutional kernel size ($kk = k^2$), $B$ is the number of output capsule types, and $D$ is the output pose dimension.

The routing process begins by applying a convolutional operation with kernel size $k$, stride $s$, and padding $p$ to transform the input poses into $\mathbf{pose}_{unfolded} \in \mathbb{R}^{b \times l \times kkA \times C \times 1}$, where $l$ represents the resulting spatial dimension. The routing coefficients are then computed as:

$$\mathbf{r} = \mathrm{softmax}(\mathbf{W}^{route}\mathbf{pose}_{unfolded}) \tag{1}$$

where $\mathbf{r} \in \mathbb{R}^{b \times l \times kkA \times B}$.

The output activations $\mathbf{a}_{out} \in \mathbb{R}^{b \times B \times h' \times w'}$ are computed by weighting the input activations with the routing coefficients:

$$\mathbf{a}_{out} = \frac{\sum_{i \in \Omega_l} \mathbf{a}_i \mathbf{r}_i}{\sum_{i \in \Omega_l} \mathbf{a}_i} \tag{2}$$

where $\Omega_l$ represents the set of input capsules in the local receptive field and $h', w'$ are the spatial dimensions after convolution.

For pose transformation, the network computes:

$$\mathbf{pose}_{out} = \frac{\sum_{i \in \Omega_l} \mathbf{r}_i \mathbf{a}_i (\mathbf{W}^{pose}\mathbf{pose}_i)}{\sum_{i \in \Omega_l} \mathbf{r}_i \mathbf{a}_i} \tag{3}$$

where $\mathbf{pose}_{out} \in \mathbb{R}^{b \times BD \times h' \times w'}$ represents the transformed pose matrices.

This formulation enables each capsule to independently determine its routing coefficients through learned parameters while maintaining the ability to model part-whole relationships. The elimination of iterative routing significantly reduces computational complexity compared to Capsule Networks with iterative routing.

While SR Caps perform competitively on standard benchmarks, their reliance on pre-learned routing network parameters hinders the network's ability to dynamically adjust routing weights based on the specific input, a key advantage of agreement-based routing approaches.

The **Stacked Capsule Autoencoder** (SCAE) Kosiorek et al. (2019) is another Capsule Network architecture which does not use supervised learning. Comprising of a Part Capsule Autoencoder (PCAE) and an Object Capsule Autoencoder (OCAE), the SCAE segments images into parts and organises these parts into coherent objects without labelled data. The model's unsupervised learning mechanism maximises the likelihood of reconstructing both the input image and inferred part poses, subject to sparsity constraints. By leveraging explicit geometric relationships between objects and their parts, SCAE achieves improved generalisation and viewpoint robustness. This approach demonstrates competitive performance in unsupervised classification tasks, particularly on MNIST and SVHN datasets, without relying on mutual information-based techniques or data augmentation. Consequently, SCAE represents a substantial paradigm shift from the conventional Capsule Network training paradigm.

## 2.2 Masked Autoencoders

Masked Autoencoders (He et al., 2021) are a specific variant of ViTs which are pretrained via a patch-specific reconstruction loss, tasking the network to reconstruct masked patches based upon the information which

can be learnt from the visible patches, this can be seen visually in figure 4. An image is first split into $N \times N$ patches of equal size and are flattened, allowing for the tokenisation of an image akin to text in a standard transformer (Vaswani et al., 2017). To mask specific areas (patches) in an image, tokens are randomly selected up to a specified percentage of the total tokens and removed from the sequence. This process eliminates the information from the feature map. The remaining visible patches are subsequently processed via a ViT. After the encoder has finished processing the visible patches, the masked patches are reinserted into the sequence, replacing the previously removed patches. The network now employs a ViT decoder to make predictions for these masked tokens utilising the attention mechanism and multilayer perceptrons within the standard ViT blocks. This process requires the network to learn how local areas correspond to their neighbouring patches by predicting the removed patches.

## 3 Masked Capsule Autoencoders

To create the MCAE we must first define how Capsule Networks can have their feature maps masked. In CNNs, this task is challenging and is typically achieved by setting specific regions of the feature map to zero. However, this method differs from the masked autoencoder (He et al., 2021). Zero masking has been demonstrated to alter the distribution of pixels in an image (Balasubramanian & Feizi, 2023), which in turn affects the performance. Therefore, in the subsequent section, we will discuss the modifications we have implemented to enable accurate masking within our MCAE.

### 3.1 Flattened Feature Map

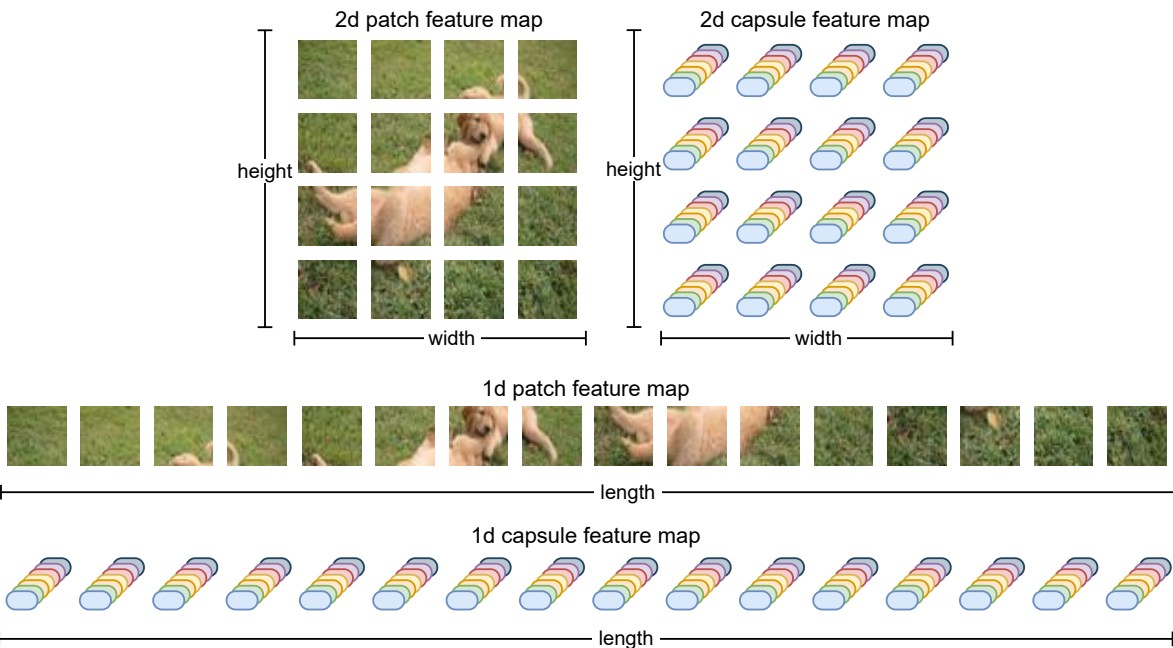

Figure 2: A visual representation of how a 2D patch feature map or capsule feature map with height and width is flattened into a 1D feature map with a length instead. At each location, there is the same amount of different capsule types, each corresponding to a different part or concept in the part-whole parse tree. The dog image used is sourced from the Imagewoof validation set (Howard, 2019b).

Vision Transformers can easily perform masking on a feature map, as patches of the image can be removed from computation by simply removing selected patches from the flattened sequence of patches after the patch embedding layer. Capsule Networks on the other hand have traditionally used a 2D feature map, which comes with the drawback that masking can only be achieved either via replacing masked regions with

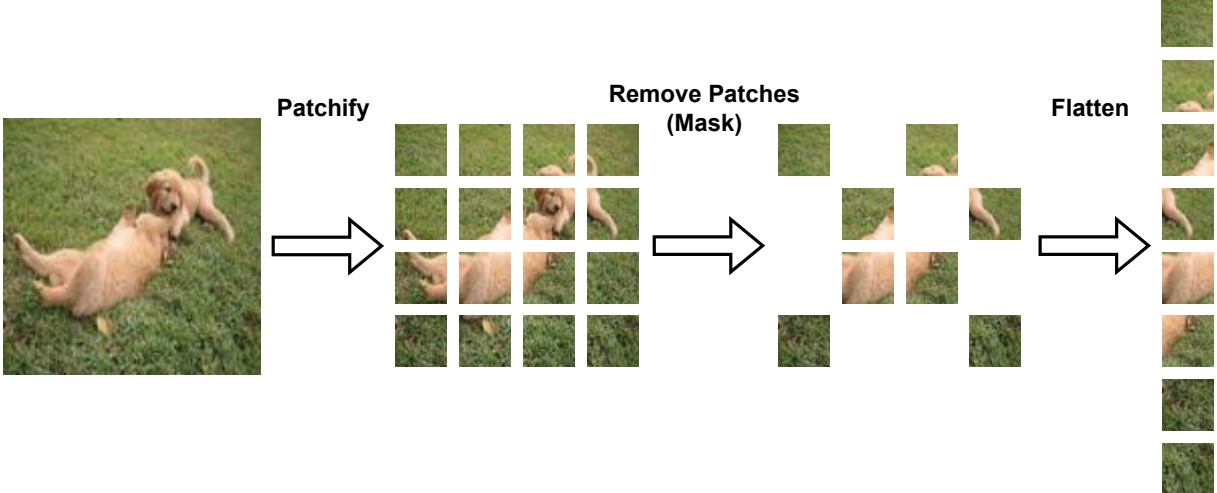

Figure 3: A visual representation of the masking process. An image is split into non-overlapping patches of $N \times N$ pixels. Randomly, a percentage, in this case, 50% of patches are removed to deprive the network of information available in these patches. The patches are then flattened into a 1D sequence of the remaining patches, ready to be processed by our encoder. The dog image used is sourced from the Imagewoof validation set (Howard, 2019b).

0's or utilising sparse operations (Woo et al., 2023), which come with their own drawbacks (Balasubramanian & Feizi, 2023; Tian et al., 2023).

Thus we propose that by flattening the 2D feature map into a 1D feature map, mimicking the design of a ViT feature map, masking can be achieved in the same way as in the Masked Autoencoder (He et al., 2021). We thus achieve masking by simply removing all capsules at a specific location along the length dimension of our feature map.

## 3.2 Building Upon Self Routing Capsule Networks

We use SR Caps Network Hahn et al. (2019) as our starting point due to its simplicity and speed, but modify its routing architecture to better suit our masked image modelling approach. The key architectural change lies in how we handle spatial relationships in routing.

The original SR Caps model uses a sliding window approach where each output capsule considers inputs from a local region $k \times k$ of the feature map. In contrast, our MCAE uses point-wise routing where capsules only route to the layer above at their exact spatial position in a flattened 1D feature map shown in 2. This creates a local parse tree for each patch position, which aims to determine the concept represented in the patch.

During pretraining, we do not include the final class capsule layer that would normally be part of a supervised Capsule Network. Instead, after the encoding stage, we introduce a learnable masked token - a shared vector of the same dimension as our capsule vectors. This masked token is inserted at all positions where patches were originally removed, returning the sequence to its original length. Following the design principles of the original masked autoencoder He et al. (2021), this same learnable placeholder is used across all masked positions and is updated during training as a parameter of the model.

The decoder then uses a fully-connected capsule layer where each capsule can route to all capsules in the subsequent layer regardless of spatial position. This is achieved by setting the kernel size to match the sequence length. While this global routing enables capsules at any position to influence the reconstruction of any masked patch, it comes with a significantly increased computational cost compared to the position-wise routing used in the encoder. However, this computational overhead only affects the pretraining phase. Once pretraining is complete, the decoder is discarded. This design choice trades higher pretraining computation

for improved network accuracy during inference, where the network also maintains the fast inference speed of SR Caps.

When we switch to finetuning, we remove this decoder architecture and replace it with a class capsule layer. This new layer averages the activations of each capsule type across all spatial positions to make class predictions, as is standard in the self routing training procedure. However, this layer is now able to leverage the improved representations required for reconstruction learned during pretraining.

### 3.3 Loss Function

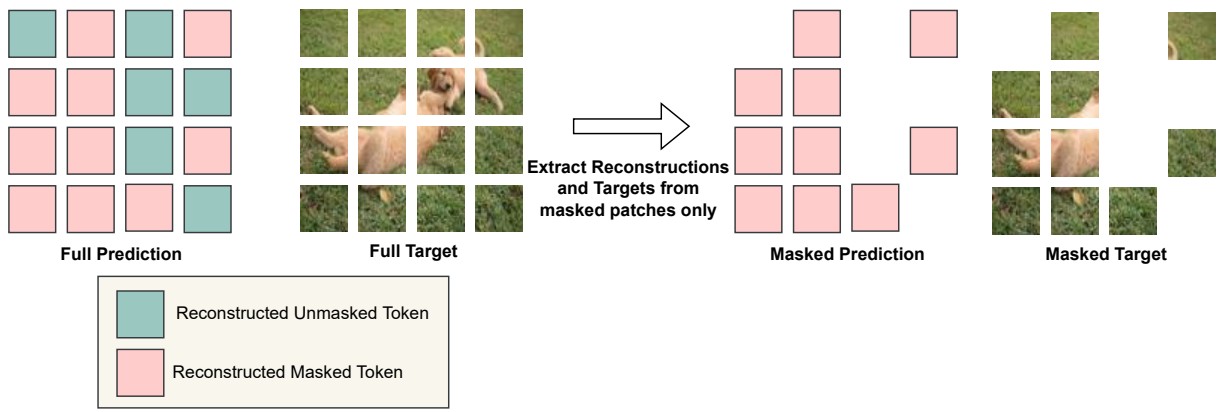

Figure 4: A visual representation of how our pretrain loss function selects patches for the loss function defined in equation 4. The dog image used is sourced from the Imagewoof validation set (Howard, 2019b).

A crucial aspect of the pretraining stage of the MCAE involves training the network to accurately reconstruct the masked portions of the input image. To achieve this, we use the Mean Squared Error (MSE) loss, which quantifies the difference between the actual pixel values of the masked patches and the predicted pixel values generated by the capsule decoder. MSE loss is defined as:

$$\text{MSE} = \frac{1}{N} \sum_{i=1}^{N} (y_i - \hat{y}_i)^2 \tag{4}$$

where $N$ represents the total number of pixels across all masked patches in the training batch, $y_i$ is the actual value of the $i^{th}$ pixel in the masked patch, and $\hat{y}_i$ denotes the predicted value from our capsule decoder for the same pixel. A visual representation of patch selection from the target and prediction can be seen in figure 4.

The MSE loss aligns with our objective to minimize the difference between the reconstructed and original patches, ensuring precise prediction of masked patch pixel values by the capsule decoder. It accentuates larger discrepancies by squaring errors, thereby pushing the model to improve on significant deviations and enhance reconstruction on each masked patch.

When finetuning for classification, the MSE loss is replaced with the cross entropy (CE) loss, defined by:

$$CE = -\sum_{i=1}^{N} \sum_{c=1}^{C} y_{ic} \log(\hat{y_{ic}}) \tag{5}$$

where $N$ is the number of samples, $C$ the number of classes, $y_{ic}$ indicates if class $c$ is correct for sample $i$, and $\hat{y_{ic}}$ is the average activation for each class $c$ of sample $i$. This loss encourages the model to activate the correct class capsules with high confidence.

### 3.4 Backbone Selection

To ensure that information is completely masked out, we replace a standard ResNet (He et al., 2016) or ConvNet backbone with a ConvMixer (Trockman & Kolter, 2022). This architecture's first layer uses a kernel size and stride of equal size, known as a patch embedding layer, allowing our feature map to contain no overlapping information. This ensures that when regions of the image are masked, information cannot be leaked through the overlapping sliding convolutional kernel.

We also provide a set of architectures with a ViT backbone. This is achieved by setting the dimension of each token's representation to *Number of Primary Capsules × Primary Capsule Embedding Dimension* allowing for an easy reshape into the primary capsule tensor dimensions. To create the activations for the primary capsules, we use a simple linear layer with sigmoid activation to ensure that the value of the activation remains between 0 and 1.

## 4 Experiments

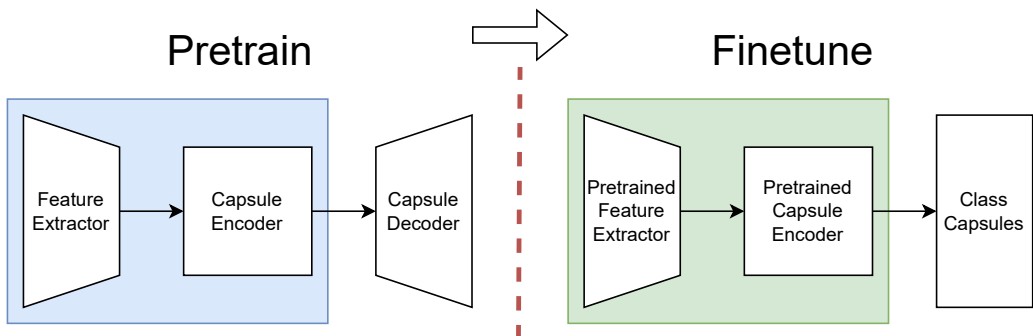

Figure 5: A visual depiction of the pretrain and finetuning components. We show how the feature extracting CNN and capsule encoder are kept from the pretrain to finetune step. The capsule decoder is discarded after pretraining and replaced with a class capsules layer that maps the capsule encoder network to a classification output.

To validate the effectiveness of our method, we conducted numerous experiments with various ablations on multiple datasets. These experiments demonstrate that masking is indeed a powerful technique for pushing the boundaries of Capsule Networks.

### 4.1 Experimental Setup

All of our experiments follow the same experimental setup. This involves to optionally pretrain the network minus the class capsules for 50 epochs, with 50% of patches removed on either 'removed patch' or 'whole image reconstruction' as a target. We then add the class capsules to our network and fully finetune it for 350 epochs, following the supervised training settings of (Hahn et al., 2019; Everett et al., 2023). A visual depiction of the elements of the components of pretraining and finetuning can be found in figure 5. All models use the SGD optimizer with default settings and the cosine annealing learning rate scheduler with a 0.1 initial learning rate.

When a validation dataset has not been predefined, we randomly split 10% of the training dataset to act as our validation dataset. The best model is tested once on the test set of our datasets, with the best model being chosen based on the epoch with the lowest validation loss.

### 4.2 Datasets

We validate our results on multiple datasets. For all of our benchmark datasets, we use the augmentation strategy proposed in (Everett et al., 2023), which aligns with the augmentations used in other capsule

papers, as we are the first to provide results on Imagenette, we define the augmentations to be the same as the augmentations for Imagewoof.

Initially, we provide a sanity check on the MNIST dataset (LeCun et al., 2010), to provide quick experimentation to ensure that our methods work at all. Next, we use both the FashionMNIST and CIFAR-10 datasets (Xiao et al., 2017; Krizhevsky et al., 2009), two datasets which are well within the abilities of a standard Capsule Network and allow us to ensure that we are not limited to the simplest of experiments. For these two datasets, the augmentation that we use is a random resized crop of 0.8 to 1.0 during training. The SmallNORB dataset (LeCun et al., 2004) allows us to ensure that we are maintaining the equivariant properties and generalisation abilities of Capsule Networks as the test set is specifically chosen to vary substantially from the train set while remaining within a similar distribution. In addition to standard classification accuracy on the SmallNORB dataset, we also follow (Hahn et al., 2019; Ribeiro et al., 2020; Hinton et al., 2018) and test our model on the novel azimuth and elevation tasks to verify generalisation capabilities. The augmentations that we use for this dataset are that we standardise and take random 32x32 crops during training. At test time, we centre crop the images to 32x32 as defined in (Ribeiro et al., 2020). Finally, we use the Imagenette and Imagewoof datasets (Howard, 2019a;b) to test our network's performance on larger, more realistic datasets. Imagenette and Imagewoof take 10 different classes from the Imagenet dataset (Deng et al., 2009). Imagenette is designed to be easily differentiable and tests our network's ability to process larger, more complex images. Imagewoof is ten classes of dogs and is designed to be more difficult to differentiate between classes due to the highly overlapping shared features. For Imagewoof and Imagenette we also resize the images to 64x64 for computational efficiency. We again use a random resize crop and colour jitter for these datasets.

### 4.3 Results

### 4.3.1 Results on Image Classification:

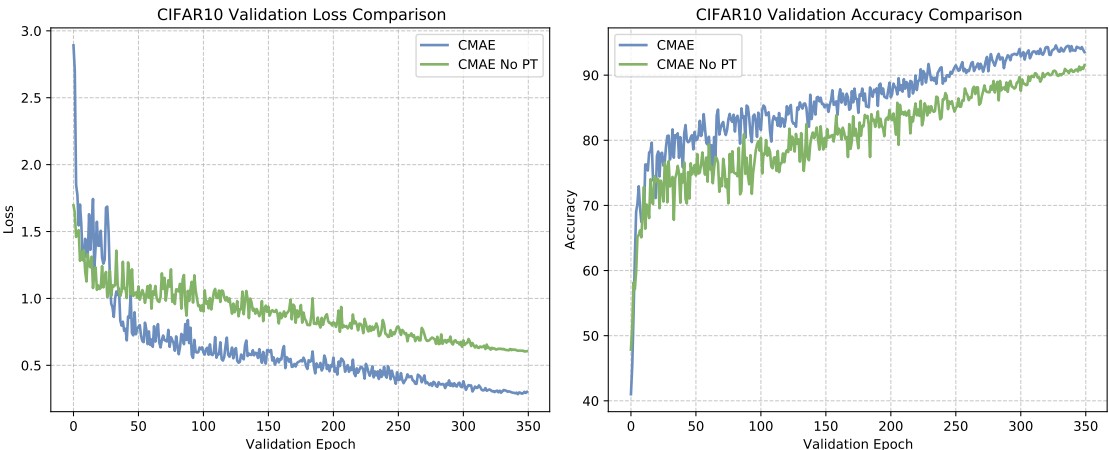

Figure 6: **Graphs comparing MCAE with vs MCAE w/o the pretraining stage**. The graphs show the validation loss (left) and validation accuracy (right) on CIFAR10 over 350 epochs.

Table 1 presents the classification results of key state-of-the-art Capsule Networks compared to our approach with no pretraining and with pretraining on the datasets proposed in our experimental design. MCAE with no pretraining is architecturally similar to SR Caps Networks, but with the 1D modification to the feature map and $1 \times 1$ kernels, along with the other required changes to the computation to allow for this. This method produces better results compared to SR-Caps, but does not achieve state-of-the-art in any dataset. However, when we apply the masked pretraining paradigm, our results improve on all datasets except MNIST. This pushes the MCAE with pretraining to be state-of-the-art for Capsule Networks in all datasets except SmallNORB, which is still dominated by Capsule Network models based on iterative routing methods. In addition to Capsule Network models, we have trained three baseline models using the exact

Table 1: Main results for a number of foundational Capsule Network models compared to both the MCAE with masked pretraining and without. We show results on the four datasets that Capsule Networks are traditionally benchmarked on, as well as providing results for the Imagenette and Imagewoof datasets which are subsets of the Imagenet dataset. Unfortunately, it is computationally infeasible to train DR, EM or VB Caps on these larger datasets due to their heavy VRAM requirements. Results for similarly sized CNNs and ViTs have been included using the results from Everett et al. (2023) and additional Imagenette results using our experimental settings for Imagewoof for fair comparison.

| | MNIST | FashionMNIST | CIFAR-10 | SmallNORB | Imagenette | Imagewoof |
|---|---|---|---|---|---|---|
| ResNet18 (He et al., 2016) | 99.1 | 89.3 | 78.4 | 89.8 | 68.9 | 52.7 |
| MobileNetv3 (Howard et al., 2019) | 99.5 | 98.7 | 77.1 | 83.3 | 62.4 | 45.9 |
| ViT-S (Dosovitskiy et al., 2020) | 99.4 | 99.3 | 80.0 | 91.1 | 70.1 | 57.3 |
| DR Caps (Sabour et al., 2017) | $99.5^{\dagger}$ | $82.5^{\dagger}$ | $91.4^{\dagger}$ | $97.3^{\dagger}$ | - | - |
| EM Caps (Hinton et al., 2018) | $99.4^{\dagger}$ | - | $87.5^{\dagger}$ | - | - | - |
| VB Caps (Ribeiro et al., 2020) | $\mathbf{99.7}^{\dagger}$ | $94.8^{\dagger}$ | $88.9^{\dagger}$ | $\mathbf{98.5}^{\dagger}$ | - | - |
| SR Caps (Hahn et al., 2019) | - | - | $92.2^{\dagger}$ | - | - | - |
| ProtoCaps (Everett et al., 2023) | $99.5^{\dagger}$ | $92.5^{\dagger}$ | $87.1^{\dagger}$ | $94.4^{\dagger}$ | 74.4 | $59.0^{\dagger}$ |
| *ResNet20 SR** (Hahn et al., 2019) | - | - | $90.0^{\dagger}$ | - | - | - |
| *SR Caps SA*** (Hahn et al., 2019) | $99.6^{\dagger}$ | $91.5^{\dagger}$ | $82.7^{\dagger}$ | $92^{\dagger}$ | 45.2*** | $32.5^{\dagger}$ |
| MCAE no PT | 99.6 | 92.1 | 91.9 | 93.1 | 73.1 | 55.9 |
| MCAE | 99.6 | **95.0** | **92.8** | 95.0 | **82.1** | **61.8** |

*The ResNet20-SR results are the performance of the ResNet20 architecture from the Self Routing Capsule Networks paper Hahn et al. (2019) which uses more augmentations, such as a random horizontal flip and normalisation, us to show the performance difference that including augmentations has on the convolutional baselines. **The SR Caps SA (Same Augmentations) results are taken from the ProtoCaps paper Everett et al. (2023) and show the performance of the SR Caps model using the same augmentations as us, note that the only overlapping dataset is Cifar10. ***The Imagenette results for SR Caps are trained using the exact experimental settings defined in ProtoCaps for their Imagewoof results. † denotes to denote any results which were taken from their original papers and thus may not use our exact experimental setting, however, care has been taken to follow conventions of training, as defined in section 4.2.*

same experimental setting as our MCAE with no pretraining regarding epochs, hyperparameters and dataset augmentations. These models were trained using the Pytorch Image Library (TIMM) Wightman (2019) and are ResNet18 (He et al., 2016), MobileNetv3 (Howard et al., 2019) and ViT-S (Dosovitskiy et al., 2020).

### 4.3.2 Backbone Choice:

Leveraging a ConvMixer backbone (Trockman & Kolter, 2022) aligns with our models' requirement of a patch embedding layer to provide non-overlapping patches of the image. ConvMixer's feature maps are by default patchified, while ViTs (Dosovitskiy et al., 2020) utilise a patch embedding layer. Prompted by this similarity, we explored this as an ablation study. Our observation reveals that ViT-based models underperform compared to those employing a convolutional backbone. Although ViT models yielded better performance than vanilla ViTs on smaller datasets, such as CIFAR-10 or SmallNORB, the overall results, shown in table 2, suggest that ConvMixers offer a more suitable architecture for the MCAE and thus have been used for our results in 1.

### 4.3.3 Reconstruction Target:

While the masked autoencoder (He et al., 2021) framework that we build upon only reconstructs masked patches, we also provide results where the reconstruction objective includes visible patches. Reconstructing based upon the whole image is inspired by DR Caps (Sabour et al., 2017) using a full image reconstruction objective along with the classification objective in order to regularise the network. The results are shown in table 3 and show that reconstructing masked patches is the best method, with reconstructing all patches providing significantly worse results.

Table 2: Results of experimentation with a ViT (Dosovitskiy et al., 2020) with depth 4 backbone compared to Capsule Networks with standard CNN backbone. The specific CNN which we use is a ConvMixer (Trockman & Kolter, 2022) of depth 4 due to its easily scalable esoteric design being based on the presumption that the image has been patchified, ensuring no information leakage of masked regions due to a sliding window of overlapping convolutional kernels. In addition, we show the mFLOPS (Total Floating Point Operations / $10^6$) to process one image through the backbone, showing that the ConvMixer is significantly more efficient. FLOPs are calculated using the FVCore library FAIR (2023).

|  | mFLOPS | Vision Transformer | mFLOPS | Conv Mixer |
|---|---|---|---|---|
| MNIST | 80 | **99.6** | 0.5 | **99.6** |
| FashionMNIST | 80 | 91.1 | 0.5 | **95.0** |
| SmallNORB | 110 | 91.4 | 2 | **95.0** |
| CIFAR-10 | 110 | 90.3 | 3 | **92.8** |
| Imagenette | 110 | 68.4 | 13 | **82.1** |
| Imagewoof | 110 | 55.4 | 13 | **61.8** |

Table 3: This table compares performance across our target datasets for MCAE pretraining based upon reconstructing both visible and masked patches versus those focusing on masked patches only. Results show equal or superior performance for models reconstructing masked patches only, across all datasets.

|  | Visible and Masked Patches | Masked Patches Only |
|---|---|---|
| MNIST | **99.6** | **99.6** |
| FashionMNIST | 88.4 | **95.0** |
| SmallNORB | 82.0 | **95.0** |
| CIFAR-10 | 84.8 | **92.8** |
| Imagenette | 45.1 | **82.1** |
| Imagewoof | 32.5 | **61.8** |

#### 4.3.4    SmallNORB Novel Viewpoint:

Table 4: Comparing novel viewpoint generalisation on the SmallNORB novel azimuth and elevation tasks (LeCun et al., 2004). Results for DR, EM and SR Caps are from (Hahn et al., 2019) and results for VB Caps are taken from (Ribeiro et al., 2020).

|  | Azimuth | | Elevation | |
|---|---|---|---|---|
|  | Familiar | Novel | Familiar | Novel |
| DR Caps | 93.1 | 79.7 | 94.2 | 83.6 |
| EM Caps | 92.6 | 79.8 | 94.0 | 82.5 |
| VB Caps | **96.3** | **88.7** | **95.7** | **88.4** |
| SR Caps | 92.4 | 80.1 | 94.0 | 84.1 |
| MCAE | 93.2 | 85.6 | 95.3 | 86.1 |

In order to verify that we retain the novel viewpoint generalisation capabilities of Capsule Networks, we use the novel azimuth and elevation tasks of the SmallNORB dataset. We replicate the experimental design of (Hahn et al., 2019; Hinton et al., 2018) and conduct two experiments. 1) Training only on azimuths in (300, 320, 340, 0, 20, 40) and test on azimuths in the range of 60 to 280. 2) Training on the elevations in (30, 35, 40) degrees from horizontal and then testing on elevations in the range of 45 to 70 degrees. In table 4 we compare our accuracy on the test set on both the seen and unseen viewpoints. We pretrain for 50 epochs and finetune for 350 epochs, the same as our best model for SmallNORB in table 1.

We do not achieve state-of-the-art results on this task, but do outperform all Capsule Networks except for VB Caps (Ribeiro et al., 2020), showing that masked pretraining does not remove the generalisation capabilities of our network.

## 5 Conclusion

We have proposed the Masked Capsule Autoencoder model, the first capsule architecture trained in a self-supervised manner. This could be a step change in the development of scalable Capsule Network models. Extensive experiments demonstrate that MCAE outperforms other Capsule Network architectures on almost all datasets, with particularly favourable results on higher-resolution images. Considering the unique and well-established advantages that Capsule Networks have in capturing viewpoint equivariance and viewpoint invariance compared with Transformers and CNNs (Ribeiro et al., 2022), our model is a step towards developing large and scalable Capsule Network models.

We would consider the drawbacks of our method to be in the fully capsule decoder. In the Masked Autoencoder paper (He et al., 2021) they state that the pretraining loss continued to decrease at the point at which they stopped pretraining at 1600 epochs. While our reconstruction loss plateaus much quicker, to the point where it does not decrease any further after the 50 epochs for which we pretrain, this indicates that there is a point at which our model has reached the best reconstructions it can achieve. While we have shown that the pretraining stage improves the maximum classification accuracy for all datasets except MNIST (due to very fine margins for quantifiable improvement), if an improved decoding mechanism can be found to benefit from additional masked pretraining, the peak classification accuracy could likely be higher. Moreover, the decoder is computationally intensive because it needs to consider the entire feature map, significantly increasing training time and VRAM requirements compared to when no finetuning is used.

## Acknowledgments

We would like to thank the University of Aberdeen's High Performance Computing facility for enabling this work and the anonymous reviewers for their constructive feedback.

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

# A Appendix

## A.1 Decoder Ablations

Table 5: Table showing ablation results of a MLP and Convolutional decoder in our MCAE model. Trained under the exact same experimental results as our MCAE model in 1. MLP models are 3 linear layer decoders of dimension num_caps * num_dim * h * w, num_caps * num_dim * h * w * 2, image_channels * img_h * img_w. Each layer has a ReLU activation function. Convolutional models use a ConvNextv2-S decoder as specified in Woo et al. (2023).

|                   | MNIST | FashionMNIST | CIFAR-10 | SmallNORB | Imagenette | Imagewoof |
|-------------------|-------|--------------|----------|-----------|------------|-----------|
| MCAE Conv Decoder | 99.6  | 91.3         | 92.1     | 93.5      | 77.3       | 57.2      |
| MCAE MLP Decoder  | 99.6  | 94.7         | 92.5     | 92.4      | 77.1       | 58.9      |

## A.2    Effect of Different Components in MCAE

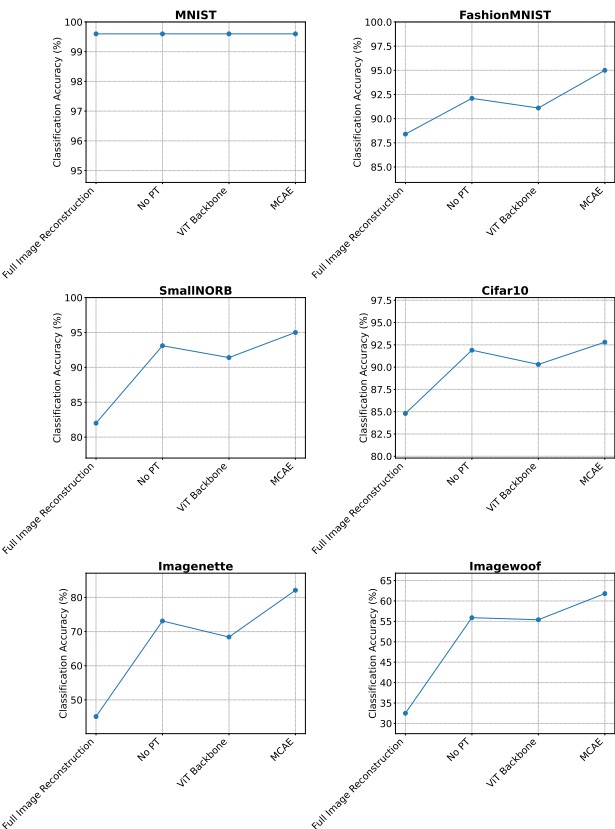

Figure 7: Graphs depicting how the top 1 accuracy changes based on different ablations of the MCAE per dataset. Full Image Reconstruction refers to a ConvMixer backbone MCAE pretrained for 50 epochs on full image reconstruction. No PT refers to a ConvMixer backbone MCAE with no pretraining epochs. ViT Backbone refers to a ViT backbone MCAE pretrained for 50 epochs on masked patch reconstruction. MCAE refers to our best-performing model which utilises a ConvMixer backbone and masked patch reconstruction. All models use the same linear SR Caps model which contains 3 layers, with 16 Capsules per layer and are finetuned for 350 epochs.

