# OpenReview forum: "Masked Capsule Autoencoders"
_TMLR — Accepted by TMLR_

### Review · Reviewer_xJ4j · 2024-07-31

**Summary Of Contributions:**

This paper introduces Masked Capsule Autoencoders (MCAE) that utilize pretraining with masked image modeling, a self-supervised learning technique, for later finetuning for image classification. It is demonstrated that pretraining a capsule network (CapsNet) in this manner enables better performance among similar CapsNet architectures. In addition to this, the MCAE is able to perform on larger, "more realistic", images where previous CapsNet architectures fail due to memory overhead. Finally, this paper investigates various architecture and training configurations of the MCAE to clearly establish which components are most beneficial for the presented results.

**Audience:**

Yes

**Broader Impact Concerns:**

None that I can think of beyond the standard ethical practice of how image classification may be used to unfairly discriminate.

**Claims And Evidence:**

Yes

**Requested Changes:**

I want to first explain my "overall" ratings below and at the outset say that I am very willing to change them throughout the review process of this paper. I will walk through my high-level reasons for saying "no" (at this point) for whether this paper satisfies the "Claims and Evidence" and "Audience" requirements of TMLR, before addressing specific changes that I feel that authors should make. Most of these opinions that I write here are not held strongly and I'm looking forward to discussing with the authors to come to an agreement. I am very comfortable with and want to change my assessment along these lines.

### Claims and Evidence

The authors make an erroneous claim at the outset of the paper and in the abstract where they state that the MCAE is the first paper to use self-supervised learning with CapsNets. There have been previous works looking into the use of unsupervised/self-supervised learning with CapsNets. I feel that the major contribution in this paper is the use of Masked Image Modeling, similar to ViT architectures. This is a neat idea within the context of CapsNets and seems to differentiate it from the prior literature. However, I think that it's important that a more thorough discussion of CapsNet architectures is needed in this paper. Specifically this paper needs to discuss how it differs from work like "Stacked Capsule Autoencoders" from Kosiorek, et al in NeurIPS 2019.

An additional concern of mine along the "Claims and Evidence" dimension is that the paper claims that MCAE can compete on equal terms with Transformers and CNNs on the included image classification benchmarks. But there are no included direct comparisons with any architectures or published methods to this end. Without empirical evidence to support this claim, I would recommend removing this statement (which would significantly affect it's "audience" estimation). Alternatively, it is necessary to include additional experimental results demonstrating this comparison.

### Audience

I am uncertain whether there is a sufficient level of interest in CapsNet architectures among the TMLR community which would warrant there being an "audience" for this paper. This is perhaps my weakest held opinion. If there was an intermediary option between "yes" and "no" for this evaluation I would've chosen it with a slight lean toward "yes". The work presented in this paper is, from my estimation, good and interesting and certainly further advances the CapsNet architecture. I just worry that there really isn't much of a community that is building upon or furthering the use of CapsNets. Does this mean that research into their use and making them better is unnecessary? Certainly not. I am uncertain however that the paper as currently written will appeal to a broader audience. I think that this could be changed by a more thorough discussion of why CapsNets are appealing alternatives to standard CNN or Transformer architectures. There is an initial attempt at this in the first paragraph of the paper but it felt somewhat insufficient to wholly motivate why continued research in CapsNets is necessary. This could also be included in the concluding section where a more thorough discussion of what advances are enabled by better performing CapsNet architectures may provide over current standards.

Some questions to consider:
> A major concern with CapsNets is that they are slow and inefficient models with a high computational overhead. How do the authors justify this? Do they demonstrate significant improvements qualitatively and quantitatively that would assuage concerns about this overhead?

## Requested changes
 - Adjust the claims made about the MCAE being the first/only unsupervised/self-supervised pretraining method for CapsNets
 - Include experimental comparisons with CNN/Transformer-based approaches.
 - Fix reference in text for Tables 2 and 3 if they are to be retained the paper.
 - Clarify what the final/intended configuration of MCAE to make the ablations more easy to follow/understand.
 - Improve discussion around the choice for the fully capsule decoder.
 - Improve discussion about how MCAE and CapsNets in general are situated in the overall ML landscape.

**Strengths And Weaknesses:**

## Strengths

This paper does a great job integrating promising techniques and modeling approaches from across ML and bring them to bear toward the improvement of CapsNets. This is they type of synthesis research that we need more of in our community.

The claim of Masked-image modeling through pre-training providing a strong improvement over prior CapsNet papers is verified in the results presented in Table 1. There is some lingering concern that MCAE isn't a total improvement (considering the SmallNORB results) but the authors do a great job discussing this and analyzing performance on SmallNORB in Section 4.3.4. Overall, I was pleased with the experimental comparisons and ablations performed in this paper. I feel that they are sufficient to demonstrate the claims made about the MCAE improving over prior CapsNet approaches.

I appreciated the work done to validate the architecture choices made with the MCAE through the ablation study performed through Section 4. As a result, I have increased confidence in the quality of the results and feel that the paper presents a nice foundation for future work to build from.

## Weaknesses

In Section 4.3.2 and Figure 6 It would be clearer if it were made more explicit that “MCAE” is built upon ConvMixer. The current sentence “results suggest that ConvMixers offer a more suitable architecure from MCAE” could be re-written into something like, “Our default setting for MCAE uses ConvMixer as a backbone. We consider an alternative construction where ViTs are used instead as a potential ablation…”

Theres is no reference to Table 2 in the text… Also, the information in tables 2 and 3 are pretty clearly identified from Figure 6. I’m not certain you need all three of these in the paper.

There are no direct comparisons to CNN or ViT based architectures.

There is little done to investigate the choice of using a "fully capsule decoder". This is discussed as drawback/weakness of the MCAE in the conclusion section. I feel that there is more to be done to justify this choice whether through an additional ablation or more focused writing/discussion beyond the single sentence at the end of the Introduction section.

---

> ### Author Response · Authors · 2024-08-30
> **Response to reviewer xJ4j part 1/3**
>
> We would firstly like to thank you for your excellent review. Below we have written a rebuttal that hopefully will alleviate your concerns, but we would love to initiate a dialogue if we have not.
>
> ### Weaknesses
>
> **"In Section 4.3.2 and Figure 6 It would be clearer if it were made more explicit that "MCAE" is built upon ConvMixer. The current sentence "results suggest that ConvMixers offer a more suitable architecure from MCAE" could be re-written into something like, "Our default setting for MCAE uses ConvMixer as a backbone. We consider an alternative construction where ViTs are used instead as a potential ablation…"**
>
> We will update our paper in both the sections you have stated and our main results table to make it clear that the ConvMixer backbone is used in our best results.
>
> **"Theres is no reference to Table 2 in the text… Also, the information in tables 2 and 3 are pretty clearly identified from Figure 6. I'm not certain you need all three of these in the paper."**
>
> We thought that figure 6 would provide a nice visual representation as to how our method performs under the different components we propose. But if the consensus from reviewers is that it is unnecessary we are happy to remove it.
>
> **"There are no direct comparisons to CNN or ViT based architectures."**
>
> We have added direct comparisons under the same experimental settings to the main results table for three models which are similar in terms of FLOPs, being ResNet18, MobileNetv3 and ViT-S.
>
> **"There is little done to investigate the choice of using a "fully capsule decoder". This is discussed as drawback/weakness of the MCAE in the conclusion section. I feel that there is more to be done to justify this choice whether through an additional ablation or more focused writing/discussion beyond the single sentence at the end of the Introduction section."**
>
> We had initially experimented with MLP and CNN based decoders. But based upon the Masked Autoencoder and ConvNextv2 models which have applied masked image modelling to transformers and CNNs using the same model as both the encoder and decoder, we felt that it would better match the paradigm by devising a fully capsule decoder. In addition, the results were also weaker for these decoders. We will add the results from our MLP and CNN based decoders to the appendix.
>
> ## Requested Changes
>
> ### Claims and Evidence
>
> **"The authors make an erroneous claim at the outset of the paper and in the abstract where they state that the MCAE is the first paper to use self-supervised learning with CapsNets. There have been previous works looking into the use of unsupervised/self-supervised learning with CapsNets. I feel that the major contribution in this paper is the use of Masked Image Modeling, similar to ViT architectures. This is a neat idea within the context of CapsNets and seems to differentiate it from the prior literature. However, I think that it's important that a more thorough discussion of CapsNet architectures is needed in this paper. Specifically this paper needs to discuss how it differs from work like "Stacked Capsule Autoencoders" from Kosiorek, et al in NeurIPS 2019. "**
>
> We accept your comment here and will adjust our claim accordingly. We view the Stacked Capsule Autoencoder to be an unsupervised work rather than self supervised, but accept that this may be confusing to the reader. We will adjust our claims to state that we are the first to implement specifically the self supervised technique of masked image modelling. In addition we will talk about previous work which differs from the standard capsule network supervised regime such as the SCAE.
>
> **"An additional concern of mine along the "Claims and Evidence" dimension is that the paper claims that MCAE can compete on equal terms with Transformers and CNNs on the included image classification benchmarks. But there are no included direct comparisons with any architectures or published methods to this end. Without empirical evidence to support this claim, I would recommend removing this statement (which would significantly affect it's "audience" estimation). Alternatively, it is necessary to include additional experimental results demonstrating this comparison."**
>
> We have added comparisons with CNNs and Vision Transformers to the table. These were previously excluded to stop the table from growing too large.

---

> ### Author Response · Authors · 2024-08-30
> **Response to reviewer xJ4j part 2/3**
>
> ### Audience
>
> **"I am uncertain whether there is a sufficient level of interest in CapsNet architectures among the TMLR community which would warrant there being an "audience" for this paper. This is perhaps my weakest held opinion. If there was an intermediary option between "yes" and "no" for this evaluation I would've chosen it with a slight lean toward "yes". The work presented in this paper is, from my estimation, good and interesting and certainly further advances the CapsNet architecture. I just worry that there really isn't much of a community that is building upon or furthering the use of CapsNets. Does this mean that research into their use and making them better is unnecessary? Certainly not. I am uncertain however that the paper as currently written will appeal to a broader audience. I think that this could be changed by a more thorough discussion of why CapsNets are appealing alternatives to standard CNN or Transformer architectures. There is an initial attempt at this in the first paragraph of the paper but it felt somewhat insufficient to wholly motivate why continued research in CapsNets is necessary. This could also be included in the concluding section where a more thorough discussion of what advances are enabled by better performing CapsNet architectures may provide over current standards."**
>
> In our opinion, Capsule Network research has stagnated because research was majority focused on finding new routing algorithms. While this is an important task, it has led to small improvements while failing to address overall issues with capsules, i.e their inability to handle more complex images. Similarly to how we have discovered new training regimes for ViTs and CNNs to improve performance on complex datasets, we are among the first to propose a new training regime for capsule networks which could potentially become the norm. Thus we think that while interest in capsules may be lower than it was in the past, our work may spark other researchers to build upon self supervised capsule networks or perhaps to reconsider whether the standard supervised training regime is the only way forward.
>
> **"A major concern with CapsNets is that they are slow and inefficient models with a high computational overhead. How do the authors justify this? Do they demonstrate significant improvements qualitatively and quantitatively that would assuage concerns about this overhead?"**
>
> There have been a number of works [1, 2] which show that Capsule Networks outperform ConvNets/ViTs on tasks which specifically require equivariance as well as invariance. However these studies are generally limited to small datasets due to the computational overhead. A recent paper has shown that these properties of capsule networks also hold on a larger dataset [3]. This shows promise that should be further investigated. In addition, there are a number of techniques addressing efficiency of computation which have not yet been applied to capsule networks from the transformer and cnn literature. If performance can be proven then we strongly believe that the trend of making models more efficient would also apply to capsules.
>
> ### Specific Requested Changes
>
> **"Adjust the claims made about the MCAE being the first/only unsupervised/self-supervised pretraining method for CapsNets"**
>
> The stacked capsule autoencoder is different from our method in that it is fully unsupervised rather than self supervised, we would consider these two learning mechanisms to be fairly different as self supervised learning aims to learn features which will be good for supervised tasks when finetuned with labels by finding training tasks which can be generated without any explicit labels, while unsupervised learning aims to solve the problem without ever needing labels.
>
> However, we do acknowledge that the SCAE did change the capsule learning methodology before us, and as such we have acknowledged their work in our related work section.
>
> **"Include experimental comparisons with CNN/Transformer-based approaches."**
>
> We have included more experimental comparisons with CNNs and a ViT model with similar compute requirements to our MCAE.
>
> **"Fix reference in text for Tables 2 and 3 if they are to be retained the paper."**
>
> We have fixed the references to tables 2 and 3 in the text. We have also moved figure 6 into the appendix as you rightfully said it shows the same information as the tables. If you believe that it should be completely removed we can also do that.
>
> **"Clarify what the final/intended configuration of MCAE to make the ablations more easy to follow/understand."**
>
> We have added in a sentence to explain that the ConvMixer backbone is the final backbone which we use.

---

> > ### Comment · Reviewer_xJ4j · 2024-09-02
> > **Please add these justifications to the main body of the paper**
> >
> > I really appreciate the effort that has been made by the authors to address my concerns. I think that the inclusion of the discussion about the comparisons between CapsNets and alternative architectures would be extremely valuable to add to the paper to more appropriately set it in the framework of the relevant literature and appeal to a broader audience, perhaps?

---

> ### Author Response · Authors · 2024-08-30
> **Response to reviewer xJ4j part 3/3**
>
> **"Improve discussion around the choice for the fully capsule decoder."**
>
> We have added to our appendix experiments with a MLP and Convolutional decoder. We show empirically that this does not work as well as the Capsule decoder.
>
> **"Improve discussion about how MCAE and CapsNets in general are situated in the overall ML landscape."**
>
> We have added additional discussion to our introduction in order to motivate why it is important that we continue to investigate modifications of capsule networks, based upon how both CNNs and ViTs have been improved iteratively over the years to go from good architectures to great architectures, we feel capsule networks have potential to do the same if their problems can be iteratively overcome.
>
> [1] Ribeiro, Fabio De Sousa, Georgios Leontidis, and Stefanos Kollias. "Capsule routing via variational bayes." Proceedings of the AAAI Conference on Artificial Intelligence. Vol. 34. No. 04. 2020.
>
> [2] Hinton, Geoffrey E., Sara Sabour, and Nicholas Frosst. "Matrix capsules with EM routing." International conference on learning representations. 2018.
>
> [3] Everett, Miles, et al. "Capsule Network Projectors are Equivariant and Invariant Learners." arXiv preprint arXiv:2405.14386 (2024).

---

> > ### Comment · Reviewer_xJ4j · 2024-09-02
> >
> > I'm not sure I saw this added discussion? I scanned the proposed edits a couple times and didn't see it.

---

> ### Author Response · Authors · 2024-09-09
> **Further changes to manuscript based upon feedback**
>
> Hello, thank you for your continued attention
>
> We have added a paragraph at the end of our introduction in response to
>
> **"I really appreciate the effort that has been made by the authors to address my concerns. I think that the inclusion of the discussion about the comparisons between CapsNets and alternative architectures would be extremely valuable to add to the paper to more appropriately set it in the framework of the relevant literature and appeal to a broader audience, perhaps?"**
>
> In addition we have added a larger discussion on the fully capsule decoder in section 3.2 (the second from last paragraph). In response to:
>
> **"I'm not sure I saw this added discussion? I scanned the proposed edits a couple times and didn't see it."**
>
> We hope that this helps to alleviate your concerns, but please let us know if there is anything else

---

> > ### Comment · Reviewer_xJ4j · 2024-09-15
> > **Yes, thank you**
> >
> > Thank you for the revisions. I will update my estimation of the paper's ability to provide evidence for its claims as well as in support of it's audience.

---

> > > ### Author Response · Authors · 2024-09-16
> > > **Thank you**
> > >
> > > Thank you for time and effort that you have put into helping us refine our paper to be the best that it can be.
> > >
> > > Signed
> > > The authors

---

### Review · Reviewer_QHq8 · 2024-10-05

**Summary Of Contributions:**

This paper proposes incorporating masked image modeling (MIM) into the training pipeline of capsule networks. Comparisons with various capsule networks on image datasets show that MIM results in modest gains for small datasets like MNIST and CIFAR-10, while leading to significant improvements on two 10-class subsets of the ImageNet dataset.

**Audience:**

Yes

**Claims And Evidence:**

No

**Requested Changes:**

Since a key metric for evaluating the proposed method and claims in this paper is image classification performance, would it be possible for the authors to provide the training and test loss/accuracy trajectories over the course of training? This would significantly aid in evaluating the results and supporting the claims made in this paper.

**Strengths And Weaknesses:**

**Strengths:**


While existing work focuses on the routing mechanisms of capsule networks, this paper introduces MIM into the training pipeline, aiming to enhance scalability to more complex datasets.


**Weakness/Clarifications:**


The authors state that capsule networks struggle to scale to more complex datasets, even with fewer parameters. However, why not increase the number of parameters to match larger models? Would that solve the scaling issue?

Additionally, the authors claim that MIM aims to address capsule networks' weaknesses by improving local representations. It is unclear how improving local representations resolves the challenge of scaling to larger datasets.

A major concern relates to evaluation: ResNet18 achieves only 78.4% on CIFAR-10 and 68.9% on Imagenette. These results seem unusually low compared to expected performance.

---

> ### Author Response · Authors · 2024-11-08
> **Response to reviewer QHq8 1/1**
>
> Thank you for reviewing our paper. We hope that the following response and the updates which we have made to our paper are able to alleviate your concerns.
>
> ## Weakness/Clarifications:
>
> **"The authors state that capsule networks struggle to scale to more complex datasets, even with fewer parameters. However, why not increase the number of parameters to match larger models? Would that solve the scaling issue?"**
>
> Unfortunately, scaling Capsule Networks is still an unsolved task. We were hoping that self supervised learning would have also allowed capsule networks to scale in the same way that ViTs or CNNs do, but we experienced the same problem as in [2] where adding more layers does not increase model performance. Thus our main contribution is to improve the results of the network without scaling, however we hope that our methodology can help improve a scaled capsule network when a technique is found to do so.
>
> **"Additionally, the authors claim that MIM aims to address capsule networks' weaknesses by improving local representations. It is unclear how improving local representations resolves the challenge of scaling to larger datasets."**
>
> Capsule Networks have been shown to have a large amount of dead capsules in [1], thus we believe that by having better local features, the network will utilise its capacity better, resulting in better scaling properties. Although we didn’t manage to achieve any better scaling properties, we did increase the overall accuracy of the network and hope that our work will be built upon to improve the overall capsule network landscape by not only unnel visioning on improving the routing algorithm, but also investigating the training regime.
>
> **"A major concern relates to evaluation: ResNet18 achieves only 78.4% on CIFAR-10 and 68.9% on Imagenette. These results seem unusually low compared to expected performance."**
>
> The ResNet18 performance is indeed lower than the highest that has been achieved with a ResNet18, but this is because we do not use the heavy augmentation regime that the state of the art models would, using techniques like Mixup etc. This is to keep our evaluations fair and in line with the rest of the capsule literature as deviating from this regime would either unfairly penalise our model against the resnet, or unfairly boost our results against other capsule network models [1].
>
> To reproduce our results on CIFAR-10 and Imagenette, you can use the TIMM library training script like so:
>
> python3 train.py torch/cifar10 –model resnet18 –dataset-download –data-dir $DATA_DIR –dataset torch/cifar10 –no-aug –img-size 32
>
> python3 train.py $DATA_DIR/imagewoof –model resnet18 –no-aug (following the advice on setting up your dataset from https://timm.fast.ai/training_scripts)
>
> ## Requested Changes:
>
> **"Since a key metric for evaluating the proposed method and claims in this paper is image classification performance, would it be possible for the authors to provide the training and test loss/accuracy trajectories over the course of training? This would significantly aid in evaluating the results and supporting the claims made in this paper."**
>
> We have added this to our paper as figure 6. We show the validation accuracy and loss of our two best runs on CIFAR10. We cannot show test loss/accuracy as we do not show our model the test set until training has finished. Please see the latest version.
>
> [1] Mitterreiter, Matthias, et al. "Why capsule neural networks do not scale: Challenging the dynamic parse-tree assumption." Proceedings of the AAAI Conference on Artificial Intelligence. Vol. 37. No. 8. 2023.
>
> [2] Everett, Miles, Mingjun Zhong, and Georgios Leontidis. "Protocaps: A fast and non-iterative capsule network routing method." TMLR (2023).

---

> > ### Comment · Reviewer_QHq8 · 2024-11-08
> >
> > Thank you for your responses. I have a couple of follow up questions.
> >
> > 1. Did you empirically verify that the improved accuracy is indeed due to better local features? This is an important claim, and its correctness should be carefully verified.
> >
> > 2. Does "–no-aug" means no augmentation is used at all, or does it include basic augmentations such as random flipping, cropping, and normalization? I ask because the accuracy reported for ResNet18 is significantly lower than expected. Could you clearly specify the augmentations used for the results in Table 1?
> >
> > In particular, you mentioned that "This is to keep our evaluations fair and in line with the rest of the capsule literature". However, I noticed that in Sect 5.1 of SR Caps (Hahn et al., 2019): "During training on CIFAR-10 and SVHN, we augment using random crop, random horizontal flip and normalization. For SmallNORB, we follow the setting of [16]; we downsample training images to 48 × 48, randomly crop 32 × 32 patches, and add random brightness and contrast. Test images are center cropped after downsampled."
> >
> > Additionally, in Table 1 of SR Caps (Hahn et al., 2019), the CNN baselines compared achieve over 90% accuracy on CIFAR-10, which is significantly higher than the results you present.
> >
> > I strongly recommend that you clarify these discrepancies, particularly with the results in Table 1, to avoid potential trust issues with the other findings in the paper.

---

> > > ### Author Response · Authors · 2024-11-08
> > > **Response to New Comment**
> > >
> > > Hello, please see below for answers to your new queries.
> > >
> > > **"Did you empirically verify that the improved accuracy is indeed due to better local features? This is an important claim, and its correctness should be carefully verified."**
> > >
> > > We tested using non-pointwise routing, which uses a 3x3 routing sliding window, the same as SR Capsules. We found that the network got considerably worse results, showing that the pointwise routing's improvement of local representations (leading to more effective pretraining via the reconstruction task) is important for our network's performance.
> > >
> > > **"Does "–no-aug" means no augmentation is used at all, or does it include basic augmentations such as random flipping, cropping, and normalization? I ask because the accuracy reported for ResNet18 is significantly lower than expected. Could you clearly specify the augmentations used for the results in Table 1?"**
> > >
> > > --no-aug means the comprehensive augmentation suite available in the TIMM library is not used. Instead it is replaced with our cifar10 augmentations, which is simply a random resized crop of 0.8 to 1.0 during training. We have provided the link to this open source library in our previous response for transparency.
> > >
> > > We state in section 4.2 that we follow the augmentation strategy defined in ProtoCaps [1] which is "CIFAR10 datasets (Krizhevsky et al., 2009) to test our network’s proficiency in classifying more complex images. For these datasets, we do not use any augmentations other than a random resized crop of 0.8 to 1.0 of the image during training.".
> > >
> > > However, we feel that the resnet performance has no influence on the reliability of our results as we still outperform the ResNet-20 from the SR caps paper. We added ResNet-18 results at the request of reviewer xJ4j in order to add context to our results outside of the field of Capsule Networks.
> > >
> > > When it comes to SmallNORB, we follow the augmentation strategy that is similar to the one you have posted above, but is not as comprehensive. They are also defined in ProtoCaps, but was originally proposed in VB Caps [2]. The exact augmentations are “we standardise and resize all images to 48x48 and take random 32x32 crops during training. At test time, we simply center crop the images to 32x32.” [2]
> > >
> > > We have added descriptions of the exact augmentations that we use for each dataset to the paper and have uploaded a new revision to reflect this.
> > >
> > > [1] Everett, Miles, Mingjun Zhong, and Georgios Leontidis. "Protocaps: A fast and non-iterative capsule network routing method." TMLR (2023).
> > >
> > > [2] Ribeiro, Fabio De Sousa, Georgios Leontidis, and Stefanos Kollias. "Capsule routing via variational bayes." Proceedings of the AAAI Conference on Artificial Intelligence. Vol. 34. No. 04. 2020.

---

> > > > ### Comment · Reviewer_QHq8 · 2024-11-08
> > > >
> > > > Thank you for your response.
> > > >
> > > > 1. Does this mean that results in Table 1 are generated using different augmentation techniques? (i.e., we know at least SR and MCAE's augmentations were different)
> > > >
> > > > 2. I understand the other reviewer’s perspective. However, the primary issue with the current version of the manuscript is the lack of any explanation for the results with ResNet, MobileNet, and ViT. This inevitably leads to two concerns: (1) questioning the reasoning behind the unusually low performance, and (2) creating an impression that the capsule networks significantly outperform these models.

---

> > > > > ### Author Response · Authors · 2024-11-08
> > > > > **Response**
> > > > >
> > > > > **”Does this mean that results in Table 1 are generated using different augmentation techniques? (i.e., we know at least SR and MCAE's augmentations were different)”**
> > > > >
> > > > > **”I understand the other reviewer’s perspective. However, the primary issue with the current version of the manuscript is the lack of any explanation for the results with ResNet, MobileNet, and ViT. This inevitably leads to two concerns: (1) questioning the reasoning behind the unusually low performance, and (2) creating an impression that the capsule networks significantly outperform these models.”**
> > > > >
> > > > > In Table 1, the results for the ResNet18, MobileNetv3, ViTs, MCAE No PT and MCAE are all directly trained by us under the exact same experimental settings as defined in section 4.2 and the ProtoCaps paper which we derive our augmentations from. We additionally train the Imagenette results for both SR Caps and ProtoCaps following the experimental setting defined in their paper. The other results in the paper are taken directly from the original papers proposing the architectures as some of these architectures do not have easily reproducible results, so we did not want to make any statement as to the accuracy of their results. Would it perhaps be helpful if we included some kind of indication next to results which are not specifically trained by us?
> > > > >
> > > > > We are not completely sure how to resolve the issue with the CNNs underperforming as we do not use augmentations, but this is the setting that capsule networks are generally trained under, so either we give the disadvantage to Capsule Networks by comparing them to CNNs/ViTs with optimal augmentation strategies or we train and evaluate under the same training setting for all architectures.
> > > > >
> > > > > We have added the ResNet20 results from SR-Caps with the augmentations to table 1, denoted by ResNet20 SR. We hope that this shows the difference that the addition of the horizontal flip and normalisation has on the results of the Cifar10 accuracy. In addition, ProtoCaps reports their own experiments with SR-Caps that use the same augmentation strategy for Cifar10 as their ProtoCaps model (and our model), so we have added this as ‘SR Caps SA (Same Augmentations) in order to show the performance of SR-Caps both with and without the added augmentations. Please let us know if this makes things clearer.

---

> > > > > > ### Comment · Reviewer_QHq8 · 2024-11-08
> > > > > >
> > > > > > Yes, it would be helpful to clearly indicate whether the results were reproduced or directly taken from other papers, along with their respective experimental settings.
> > > > > >
> > > > > > To clarify, the issue is less about giving advantages or disadvantages to one approach over another. Rather, it’s about clearly and accurately stating the experimental settings to ensure transparency.

---

> ### Author Response · Authors · 2024-11-08
> **Response**
>
> We have uploaded a further revision to the paper. We have added indications on any result that was taken from another paper and thus not trained by us and added a warning in the caption that there may be discrepancies in the training procedure compared to ours.
>
> In addition, we make it clear in section 4.3.1 how our baseline CNN and ViT models are trained and the training procedure that we used to train them aligns exactly with our proposed model.
>
> Finally, we would like to sincerely thank you for the effort and attention to detail you have put in for this review, in order to best improve our paper.

---

### Review · Reviewer_iLVd · 2024-10-29

**Summary Of Contributions:**

The paper introduces ideas from self-supervised learning as in masked autoencoders to the capsule architecture. They train an encoder-decoder architecture with patches as input on a reconstruction loss. The major claim of the paper is that this unsupervised training + supervised finetuning can unlock capsule networks for high-resolution images and complex datasets. Results are shown comparing to other Capsule Network architectures on traditional small datasets (MNIST, CIFAR-10, SmallNORB, FashionMNIST) and more complex datasets such as ImageNette and ImageWoof

**Audience:**

No

**Claims And Evidence:**

No

**Requested Changes:**

Here a list of required changes which are needed for acceptance.

## Technical detials

* The paper builds on top of Self-Routing Capsule Networks but it is not self-contained. The last paragraph of Section 2.1 is dedicated to this but I had to refer to (Hahn et. al 2019) to understand. The paper can use precise mathematical notation to describe the inputs, outputs, pose matrix and route matrix, activations with the precise dimensions. A good inspiration for this is the second paragraph of Section 4.2 in (Hahn et.al 2019).
* In Section 3.2, the paper dives deeply into the intricacies of their modifications without describing the overall architecture first. The paper can first start with an overall description of the architecture used in Hahn et al, and then describe their proposed modifications to their architecture. Concretely, there are terms like “rather than merging local capsules within a HxW sliding kernel, we simply use a 1x1 region and route the capsules to the upper location in the same region.”, which are difficult to parse.
* Why did the paper start with SR CAPS instead of ProtoCAPS, since ProtoCAPS has better performance? This decision needs a bit more justification.
* “Resinsert a masked capsule placeholder” – What is the precise representation of the mask?
* The major idea of the margin-based loss is that the length of the vector can encode the probability of an object being present and is more aligned with capsule networks. See Section 3 of https://arxiv.org/pdf/1710.09829. The paper replaces the standard margin loss used in capsule networks with a cross-entropy loss. What is the rationale behind this?
* ConvMixer is composed of depthwise (sliding) convolutions at every layer. So the inputs to these are assumed to be spatially congruent, that is a 3-D Tensor (HWC). Throwing away patch information as done in Figure 1, removes this spatial congruence. So it’s quite unclear how authors integrated masking into the ConvMixer architecture.
* The architecture of the capsule decoder is not clearly explained in the paper.

## Experiments:

* What is the resolution of ImageNette and ImageWolf? Is it 224x224
* The ablation in Table 2 is quite difficult to assess without some numbers without  some sort of control such as wall-time or flops.
The paper claims that (MCAE no PT) is very similar to SR CAPS apart from the pretraining strategy. So it makes sense that the results between MCAE no PT and SR CAPS very similar on MNIST, Fashion MNIST, CIF. However, there exists a huge performance difference on ImageNette and ImageWoof between MCAE no PT and SR CAPS. Why is the case?
* Please bold the best results in all tables.
* The paper states that “but also a 9% improvement compared to purely supervised training”. The claim should be more specific. On which dataset and comparing to which “supervised training”?

## Writing


I found some sentences to be unreadable and unnecessary complex. Please split them into smaller, legible sentences.

* Each of these capsules has a corresponding activation value between 0 and 1 which represents how strongly the network believes the concept which the capsule represents is present at the location in the feature map.
* The capsules in the lowest layer can be thought of as the most basic parts which could be a part of any of the end classes, thus are denoted as the primary capsules, signifying that they are the base parts of the parse tree.
* This feature map which now contains both encoded capsule representations and a random noise-masked capsule representation is now fed through a capsule layer which considers all capsules at all locations in the lower layer when creating the pose vector and activation values of all capsules at all locations in the higher layer, meaning that the encoded capsules can predict the values of the masked regions.

**Strengths And Weaknesses:**

## Strengths:

* The paper has interesting ideas combining self-supervised learning with capsule networks.
* The paper archives competitive results as compared to other capsule network architectures.
* While a lot of algorithms have been proposed on how to improve the routing algorithm, this paper focuses on better and more improved training strategies.


## Weaknesses:

* The paper skips a lot of technical details regarding the actual architecture so it's quite difficult to judge the correctness of the architecture. Imho for acceptance, some parts of the paper requires a rewrite and explain the technical details
* Some of the experiment results also need to be clarified.

---

> ### Author Response · Authors · 2024-11-08
> **Response to reviewer iLVd 1/2**
>
> Thank you for reviewing our paper. We have made numerous changes to our paper in response to your feedback. Below we have responded to all of your points to hopefully alleviate any concerns you have.
>
> ### Weaknesses:
> **"The paper skips a lot of technical details regarding the actual architecture so it's quite difficult to judge the correctness of the architecture. Imho for acceptance, some parts of the paper requires a rewrite and explain the technical details."**
>
> We have rewritten several parts of the paper in response to your requested change. But if you feel that any other parts of the paper need a rewrite please indicate so so that we can improve our papers presentation.
>
> ### Requested Changes:
>
> ## Technical Details:
>
> **"The paper builds on top of Self-Routing Capsule Networks but it is not self-contained. The last paragraph of Section 2.1 is dedicated to this but I had to refer to (Hahn et. al 2019) to understand. The paper can use precise mathematical notation to describe the inputs, outputs, pose matrix and route matrix, activations with the precise dimensions. A good inspiration for this is the second paragraph of Section 4.2 in (Hahn et.al 2019)."**
>
> We have expanded our explanation of the SR notation. Previously we thought that this would be best kept brief, but based upon reviews have expanded it so that understanding of the SR paper is not required to read and understand our work.
>
> **"In Section 3.2, the paper dives deeply into the intricacies of their modifications without describing the overall architecture first. The paper can first start with an overall description of the architecture used in Hahn et al, and then describe their proposed modifications to their architecture. Concretely, there are terms like “rather than merging local capsules within a HxW sliding kernel, we simply use a 1x1 region and route the capsules to the upper location in the same region.”, which are difficult to parse."**
>
> As said above, we have improved our explanation of our method and updated the paper accordingly. The latest version of the paper has the differences highlighted in blue, please let us know if there is any further paint points in understanding..
>
> **"Why did the paper start with SR CAPS instead of ProtoCAPS, since ProtoCAPS has better performance? This decision needs a bit more justification."**
>
> We use SRCaps as it has become the standard for a performant non iterative algorithm that is easy to adapt and plugin to other algorithms.  Especially when working with capsules for self supervised learning [3]. Nevertheless, as it happens with CNNs (ResNet baseline backbones), one can utilise other capsule routing approaches for all intents and purposes, but that would not interfere with the core premises of the self-supervised formulation we are proposing, i.e. Masked Capsule Autoencoders.
>
> **"“Resinsert a masked capsule placeholder” – What is the precise representation of the mask?"**
>
> The masked capsule placeholder is simply a learnable vector of the same length as the other capsule vectors. It is reinserted at the position that the removed capsules were. The placeholders are the same at all masked locations and are learnable based upon the design of the original masked autoencoder. We have made this clearer in our paper too.
>
> **"The major idea of the margin-based loss is that the length of the vector can encode the probability of an object being present and is more aligned with capsule networks. See Section 3 of https://arxiv.org/pdf/1710.09829. The paper replaces the standard margin loss used in capsule networks with a cross-entropy loss. What is the rationale behind this?"**
>
> Margin loss was used in the original dynamic routing paper [3], but the authors' subsequent work [4] moved to a different loss function (spread loss). SRCaps uses a cross-entropy loss indeed, but most importantly, VarCaps [5] explored both the spread loss and a negative log-likelihood/cross entropy loss, ending up adopting the latter based on performance. The same occurred with the ProtoCaps paper [6].
>
> In addition, the SR paper that we build our method upon uses the cross entropy loss. Therefore, to ascertain consistency and given the empirical evidence and the properties of cross entropy, we maintain the use of it throughout.

---

> ### Author Response · Authors · 2024-11-08
> **Response to reviewer iLVd 2/2**
>
> **"ConvMixer is composed of depthwise (sliding) convolutions at every layer. So the inputs to these are assumed to be spatially congruent, that is a 3-D Tensor (HWC). Throwing away patch information as done in Figure 1, removes this spatial congruence. So it’s quite unclear how authors integrated masking into the ConvMixer architecture."**
>
> By zeroing out the input data in the masking process before it is fed to the backbone (in this case ConvMixer), the masked (zeroed out) patches are never seen, even though they are (inefficiently) processed by the backbone, thus have no influence but also requiring no architectural design changes. And yes, we do remove some spatial congruence, but this doesn’t cause any major issues in the models convergence while remaining a simple isotropic backbone as per our requirements.
>
> ## Experiments:
>
> **"What is the resolution of ImageNette and ImageWoof? Is it 224x224?"**
>
> We experimented with 224x224 but found that 1. It took too long to process and that 2. We could get almost identical results with resizing to 64x64. So our final imagenette and imagewoof experiments are completed on 64x64 resizings. We have updated the paper to make this clear.
>
> **"The ablation in Table 2 is quite difficult to assess without some numbers without some sort of control such as wall-time or flops. The paper claims that (MCAE no PT) is very similar to SR CAPS apart from the pretraining strategy. So it makes sense that the results between MCAE no PT and SR CAPS very similar on MNIST, Fashion MNIST, CIF. However, there exists a huge performance difference on ImageNette and ImageWoof between MCAE no PT and SR CAPS. Why is the case?"**
>
> The 1x1 kernel size in the routing that we use had better results than when we tried to use any other sizes. Additionally, the convmixer has better results than a resnet. The combination of both of these factors likely leads to the better results. The reason that the difference is larger on the datasets with higher resolution images is likely due to overcoming the issue of the network being distracted by background that was found to occur in capsule networks. [1].
>
> In addition, we have added the gFLOP count of our two backbones to table 2 on each dataset to show the computational performance of the two backbones.
>
> **"Please bold the best results in all tables."**
>
> We have done this.
>
> **"The paper states that “but also a 9% improvement compared to purely supervised training”. The claim should be more specific. On which dataset and comparing to which “supervised training”?"**
>
> We have clarified this in our abstract, and that it is a 9% improvement over our own method when trained in a supervised manner.
>
> ## Writing:
>
> **"I found some sentences to be unreadable and unnecessarily complex. Please split them into smaller, legible sentences.**
>
> **Each of these capsules has a corresponding activation value between 0 and 1 which represents how strongly the network believes the concept which the capsule represents is present at the location in the feature map.**
>
> **The capsules in the lowest layer can be thought of as the most basic parts which could be a part of any of the end classes, thus are denoted as the primary capsules, signifying that they are the base parts of the parse tree.**
>
> **This feature map which now contains both encoded capsule representations and a random noise-masked capsule representation is now fed through a capsule layer which considers all capsules at all locations in the lower layer when creating the pose vector and activation values of all capsules at all locations in the higher layer, meaning that the encoded capsules can predict the values of the masked regions."**
>
> We have rectified all of these circumstances that you have pointed out, please let us know if there are any other major pain points in our writing.
>
> [1] Xi, Edgar, Selina Bing, and Yang Jin. "Capsule network performance on complex data." arXiv preprint arXiv:1712.03480 (2017).
>
> [2] Everett, Miles, et al. "Capsule Network Projectors are Equivariant and Invariant Learners." arXiv preprint arXiv:2405.14386 (2024).
>
> [3] Sabour, Sara, Nicholas Frosst, and Geoffrey E. Hinton. "Dynamic routing between capsules." Advances in neural information processing systems 30 (2017).
>
> [4] Hinton, Geoffrey E., Sara Sabour, and Nicholas Frosst. "Matrix capsules with EM routing." International conference on learning representations. 2018.
>
> [5] Ribeiro, Fabio De Sousa, Georgios Leontidis, and Stefanos Kollias. "Capsule routing via variational bayes." Proceedings of the AAAI Conference on Artificial Intelligence. Vol. 34. No. 04. 2020.
>
> [6] Everett, Miles, Mingjun Zhong, and Georgios Leontidis. "Protocaps: A fast and non-iterative capsule network routing method." TMLR (2023).

---

> > ### Author Response · Authors · 2024-11-14
> > **Thank you**
> >
> > Dear Reviewer,
> >
> > We hope we have addressed all your concerns, but if there is anything else you would like us to clarify please let us know.
> >
> > Thank you again for taking the time to review our work.
> >
> > Kind Regards,
> >
> > The Authors

---

### Author Response · Authors · 2024-12-03
**Outcome**

Dear AC and Reviewers,

Many thanks once again for helping us improve our manuscript through three rounds of revisions.

We would appreciate an update as to when we should expect to hear from you regarding the outcome.

Kind Regards,

Authors

---

### Decision · Action_Editor_8W8j · 2024-12-18

**Recommendation:** Accept with minor revision

**Comment:**

The idea is interesting enough to deserve publication, even if there remains a bit of skepticism on the reviewer's side.

**Audience:**

Although Capsule autoencoders remains historically a niche topic, it is potentially of broad interest: it relies on small-scale architectures, which is important for sustainability, a major concern for AI. So, overall, I think that the paper can reach  a wide audience.

**Claims And Evidence:**

There is a clear consensus among reviewers that the paper has improved and presents strong arguments, which raises interest on Masked Capsule Autoencoders. The crucial aspect is that the main claims of the paper are well justified, which calls for acceptance according to TMLR criteria. There is still an issue with the claim that the method "outperforms purely supervised training " in the abstract
This being said, all reviewers agree that the writing is quite poor, and in particular that the description of the method remains hard to understand. So, it is absolutely necessary that the authors improve the writing.
Please consider comments issed by reviewer iLVd.
Here are some additional comments:
* Please break long sentences (there remains a lot of them) into small pieces.
* A sentence like "These proposals are based upon how both CNNs and ViTs have seen significant improvements relating to better
training regimes with little to no architectural changes, thus rather than solely focusing on improving the routing algorithm, we instead show that capsule networks can be improved via training modifications instead." p.2 is unreadable
* p.1 "the network believes" does not make sense
* p.2 bullet points use the same tense in all items
* p.3 "that could be combined to form..." maybe rather "can"
* "making them the foundation of the parse tree." please rephrase
* p.4 "SCAE thus represents a significant paradigm shift from the standard capsule network training regime." please rephrase
* p.5 ". This process requires the network to learn how local areas might correspond to their neighbouring patches by predicting the removed patches."
* p.5 "and thus effecting results." WDYM?
* p.9 missing reference
* caption of Fig.6 is poor. Please make a proper figure caption
* p.10 "t. However,
when we apply the masked pretraining paradigm, our results improve on all datasets except MNIST, pushing
the MCAE with pretraining to be state-of-the-art for Capsule Networks in all datasets except SmallNORB,
which is still dominated by iterative routing methods." please rephrase
* Table 2 use mFLOPS instead of gFLOPS

---

> ### Author Response · Authors · 2024-12-24
> **Thank you**
>
> Dear Action Editor,
>
> Thank you for communicating this excellent outcome to us. We are now preparing the camera-ready version, incorporating all the points mentioned in your decision.
>
> Many thanks.
>
> The authors.

---

> ### Author Response · Authors · 2024-12-28
> **Updated Version**
>
> Hello
>
> We have taken your feedback into account and revised the writing of the paper taking into account the areas which you pointed to specifically that needed fixing, but also fixing some other instances in the spirit of your advice. This has been updated on openreview now.
>
> If there is any further issues with our paper, please let us know.
>
> Thanks
> Authors

---

> > ### Comment · Action_Editor_8W8j · 2025-01-05
> > **The updated version looks good**
> >
> > The formulation concerns I raised have been addressed.